# Reactivity of He with ionic compounds under high pressure

Zhen Liu[1,2,3], Jorge Botana[1,3], Andreas Hermann [4], Steven Valdez[3], Eva Zurek[5], Dadong Yan[2], Hai-qing Lin[1] & Mao-sheng Miao[1,3]

Until very recently, helium had remained the last naturally occurring element that was known not to form stable solid compounds. Here we propose and demonstrate that there is a general driving force for helium to react with ionic compounds that contain an unequal number of cations and anions. The corresponding reaction products are stabilized not by local chemical bonds but by long-range Coulomb interactions that are significantly modified by the insertion of helium atoms, especially under high pressure. This mechanism also explains the recently discovered reactivity of He and Na under pressure. Our work reveals that helium has the propensity to react with a broad range of ionic compounds at pressures as low as 30 GPa. Since most of the Earth's minerals contain unequal numbers of positively and negatively charged atoms, our work suggests that large quantities of He might be stored in the Earth's lower mantle.

[1] Beijing Computational Science Research Centre, Beijing 100193, China. [2] Department of Physics, Beijing Normal University, Beijing 100875, China. [3] Department of Chemistry and Biochemistry, California State University Northridge, Northridge, CA 91330-8262, USA. [4] Centre for Science at Extreme Conditions and SUPA, School of Physics and Astronomy, The University of Edinburgh, Edinburgh EH9 3FD, UK. [5] Department of Chemistry, State University of New York at Buffalo, Buffalo, NY 14260-3000, USA. Correspondence and requests for materials should be addressed to M.-s.M. (email: mmiao@csun.edu)

The noble gas (NG) elements, such as He, Ne, Ar, Kr, and Xe, were believed not to react with other elements for decades, due to their stable closed shell electron configuration. Pauling[1] predicted that Kr and Xe may react with F and O, which was proved by Bartlett[2] who found the first NG compound, the ionic $Xe^+[PtF_6]^-$. Since then, numerous NG compounds have been synthesized, both in molecular and solid form[3–9]. Electronic structure calculations have predicted many more[10–18]. Meanwhile, the modification of external conditions such as pressure has led to the successful formation of yet different classes of NG compounds[19–25]. In most of these compounds, NG elements are oxidized and form chemical bonds by sharing their closed shell electrons.

It is no coincidence that much of the recent progress on NG chemistry has been made in the area of high pressure, especially regarding unusual bonding features. This is due to the fact that high external pressure can drastically alter the chemical properties of elements[26–28]. Recent theoretical studies showed that Xe becomes easier to oxidize under high pressure; for example, Xe can form stable compounds with oxygen[18,29]. Even though these compounds have been found at ambient conditions, they are only metastable. Under pressures as high as those in the Earth's core, Xe can even be oxidized by Fe and form stable Fe-Xe compounds[30]. In contrast to the above studies, a recent investigation demonstrated that NG elements can also become oxidants and gain electrons while forming compounds with elements with low ionization energies such as alkali and alkaline earth metals[31,32]. In these compounds, NG atoms are negatively charged and play the role of the anions. It has also been revealed that high pressure promotes the formation of Xe-Xe covalent bonds in $Xe_2F$ compounds[33]. Furthermore, compounds formed between NG elements[19,34,35] and with other closed shell systems have been reported: notably diatomic gases like $Xe-H_2$[36] and $Xe-N_2$[37] and closed shell molecules like $Xe-CH_4$[38]. Many NG elements are found or are predicted to form weakly interacting host-guest hydrates or clathrates[39–42]. In contrast to other compounds, these phases are bound by van der Waals forces.

Under ambient conditions, only the heavier NG elements Xe and Kr and, to some extent, Ar, are found to be chemically reactive. Remarkably, Dong et al.[43] reported recently in a combined experimental and computational study that mixtures of sodium (as well as its oxide) with helium can be stabilized at high pressure. A detailed electronic structure analysis of the resulting compounds $Na_2He$ and $Na_2OHe$ showed that He does not lose electrons nor form any chemical bonds. It is important to notice that the $Na_2He$ compound can be regarded as a high-pressure electride of the form $Na^+_2E^{-2}He$, where E represent the interstitial sites (quasi-atom) hosting a pair of electrons. Note that Sun et al.[44] proposed from calculations that He can react with many ionic alkali oxide or sulfide compounds under high pressure. A very recent work by Liu et al.[45] noticed the ability of He to form stable compounds with water molecules at high pressures. The origin of the stability of all the He-containing compounds above is not well understood[46].

Here we propose that helium has a general propensity to react with ionic compounds that contain an unequal number of cations and anions, e.g., $A_2B$ or $AB_2$. Such compounds have large Coulomb repulsive interactions between the majority ions (cations or anions), which leads to two effects that favor reaction with helium. First, in the lower pressure range, these repulsive interactions prevent the formation of close-packed structures, thus leaving room for the insertion of helium atoms; this means that the reaction with helium can potentially be stabilized due to the large gain in $PV$ term (compression work). More importantly, with increased pressure, the Coulomb repulsion becomes even stronger. The presence of He can then, second, keep the majority ions farther apart and therefore lower the Madelung energy. We will examine a series of example systems and show that the

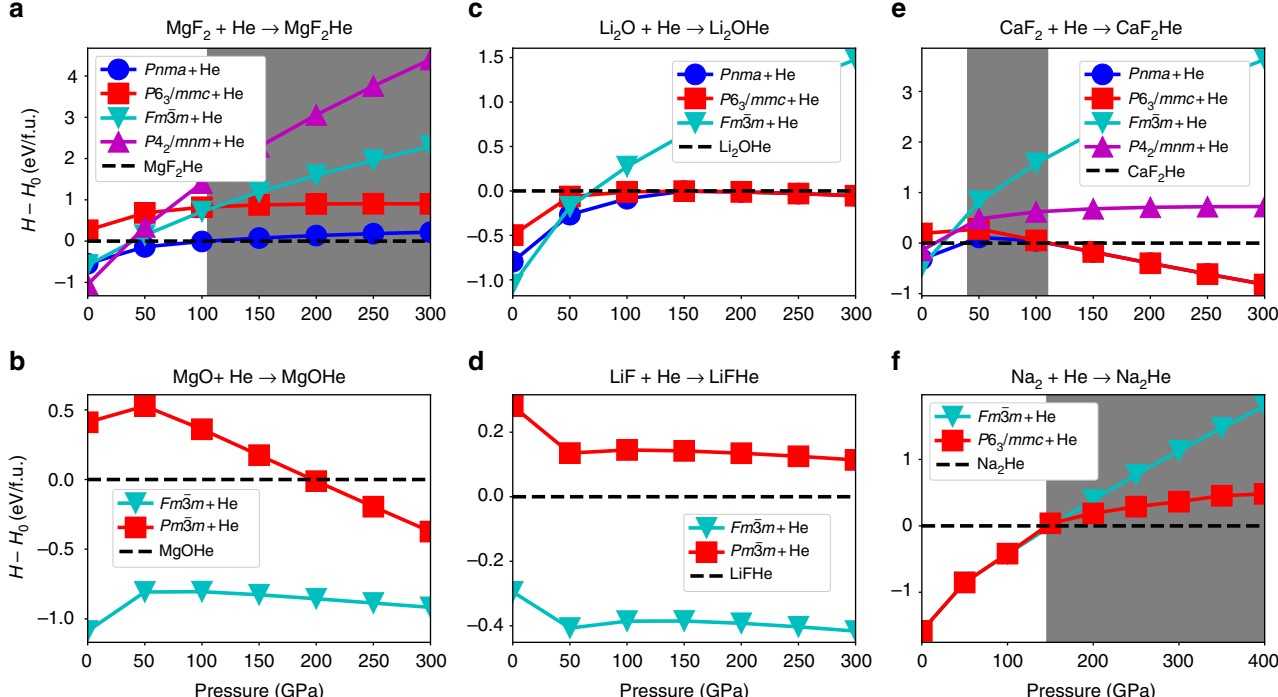

**Fig. 1** The enthalpy difference between A-B + He and A-BHe. Calculations, between helium and **a** $MgF_2$, **b** MgO, **c** LiF, **d** $Li_2O$, **e** $CaF_2$, and **f** Na, are plotted as a function of pressure. The pressure range in **a**–**e** is 0–300 GPa, and in **f** is 0–400 GPa. The dashed lines refer to the enthalpy of the He-inserted compounds. When a solid line is above the dashed line, the corresponding structure of the ionic compound is unstable relative to the He-inserted compound. Shaded areas present the pressure intervals of stable He insertion

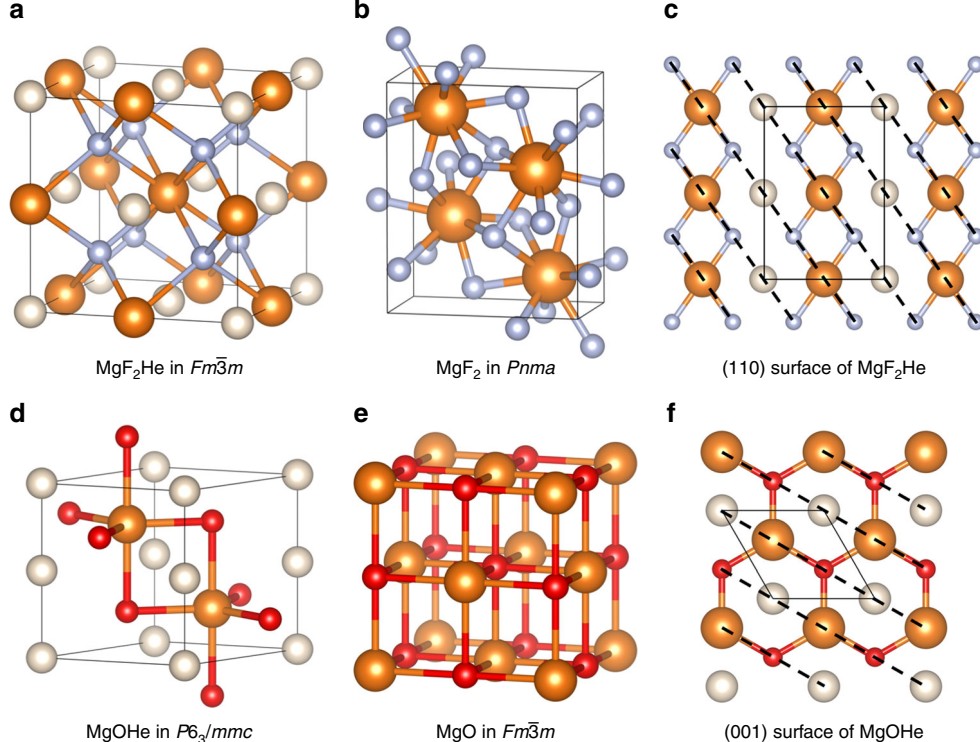

**Fig. 2** Exemplary structures A-B and A-BHe compounds. **a** $MgF_2He$-$Fm\overline{3}m$, **b** $MgF_2$-$Pnma$ at 300 GPa, **c** a (110) plane in $MgF_2He$ (see text), **d** MgOHe-$P6_3/mmc$ at 300 GPa, **e** MgO-$Fm\overline{3}m$, and **f** top view of MgOHe-$P6_3/mmc$. He (Mg, O, and F) atoms are shown as white (orange, red, and blue) spheres

combination of the two effects, namely the $PV$ and the Madelung energies, favors reactions between helium and various ionic compounds, sometimes at quite moderate compression. For number-balanced ionic compounds (chemical formula AB), the above arguments do not apply and we show that indeed helium does not react with several prototypical compounds. Through detailed energy analyses, we find that the eventual stabilities of the He (and Ne)-inserted ionic compounds depend on the balance of the above driving forces and the factors that counteract them. The reaction of He with a large number of ionic compounds shows very intriguing behavior, yet it can be explained within the framework of our theory. Our work reveals that chemically inert elements such as He can become reactive and form new compounds under pressure without the formation of any local chemical bonds.

The reactivity of He with ionic compounds may have significance in geoscience. Earth has a finite supply of helium; and due to the light weight of these atoms, they tend to escape into space. It is therefore of significant interest whether mantle materials could store large quantities of helium. Previously, the miscibility of helium in the mantle has been considered very low due to the hitherto assumed inertness of the element. However, as shown by our work, helium tends to insert into the lattices of ionic compounds with unequal cation and anion numbers at high pressure—which is a feature shared by most of the minerals in the Earth's mantle, indicating that they may store considerable amounts of helium. Of course, our calculations apply to the ground state, and the effect of elevated temperatures, inevitable inside the mantle, needs to be addressed. This is beyond the scope of current work and will be investigated later. However, our results, which will be presented in a follow-up study, are in line with recent laboratory experiments that discovered significant uptake of He in $SiO_2$ glass as well as cristobalite[47–49], a high-pressure polymorph of quartz, in the pressure range 10–20 GPa.

## Results

**Reactivity of helium with ionic compounds**. In order to test our theory, we chose four ionic compounds $MgF_2$, MgO, $Li_2O$, and LiF, and studied their reactivity with He under high pressure. These four compounds represent ionic compounds of $AB_2$ type, AB type with ±2 charge, $A_2B$ type, and AB type with ±1 charge, respectively. $CaF_2$ was also included in our study as it would reveal an important opposing mechanism caused by the occupation of the outer-shell $d$ orbitals under pressure. For comparison, we also further investigated the reaction of Na with He, which can be viewed as the interaction of the ionic compound $Na_2E$ with He. We first searched for the most stable structures of these compounds with and without insertion of He atoms under pressures from 0 to 300 GPa. Then, the enthalpy change for the inclusion of He in these compounds is calculated in the same pressure range. The enthalpy differences for the reaction A-B + He → A-BHe were calculated as follows:

$$\Delta H^r = \left(\Delta H^f_{A-B} + \Delta H^f_{He}\right) - \Delta H^f_{A-BHe} \qquad (1)$$

Note the difference between $\Delta H^r$ here and the reaction enthalpy. A positive $\Delta H^r$ corresponds to an exothermal reaction, or a thermodynamically stable A-BHe compound. The results of $\Delta H^r$ as function of pressure are shown in Fig. 1. Since the ionic compounds may undergo structural changes under increasing pressure, several $\Delta H^r$-$P$ curves corresponding to different structures are shown. In contrast, the most stable structure of each A-BHe compound remains the same throughout the pressure range.

Let us first compare the results of $MgF_2$-He and MgOHe. The former compound has twice as many anions ($F^-$) as cations ($Mg^{2+}$); whereas in the latter, their numbers are equal. As shown in Fig. 1a, a 1:1 mixture of $MgF_2$ and He will become stabilized as a ternary compound, $MgF_2He$, between 100 and 150 GPa (at an

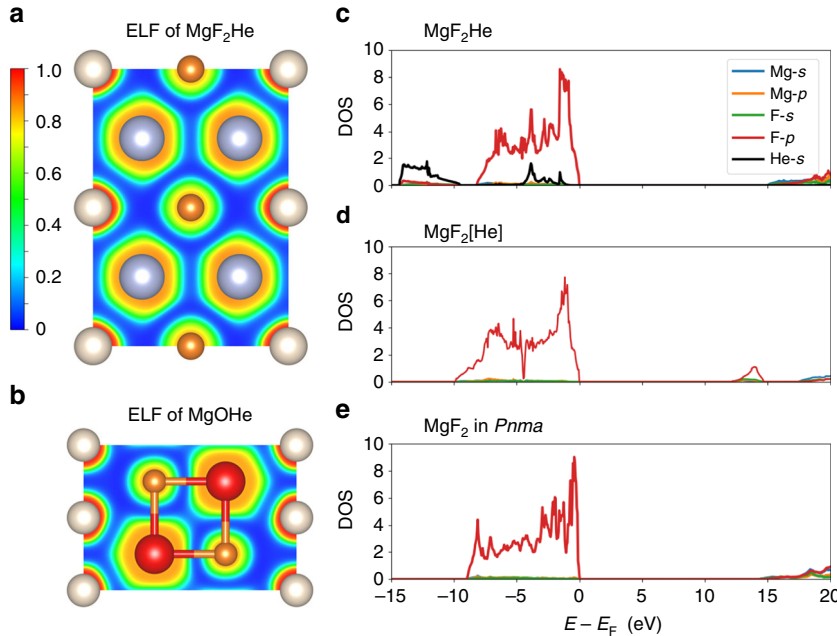

**Fig. 3** Electronic structures of $MgF_2He$ and MgOHe. **a** ELF in the (110) plane of $MgF_2He$-$Fm\bar{3}m$. **b** ELF in the (100) plane of MgOHe-$P6_3/mmc$. **c** The electronic PDOS of $MgF_2He$ at 300 GPa, **d** PDOS of hypothetical compound $MgF_2[He]$. **e** PDOS of $MgF_2$-$Pnma$ at 300 GPa

interpolated value of 107 GPa). At ambient pressure, $MgF_2He$ is 0.25 eV/atom higher in enthalpy than the constituents $MgF_2$ and He. However, at 300 GPa, $MgF_2He$ is about 0.05 eV/atom lower in enthalpy (Fig. 1a). We considered adding more He by calculating the stability of $MgF_2He_2$ compounds as well (Supplementary Figure 1 and Supplementary Note 1). Although their enthalpy decreases by a small amount from 0 to 50 GPa, it then increases again at higher pressure, and ultimately no stable $MgF_2He_2$ could be found. Therefore, $MgF_2$-He compounds can only be stabilized within a limited composition range. In contrast to $MgF_2$-He, MgOHe cannot form any stable compound at any composition ratio. For the 1:1 compound MgOHe, the enthalpy decreases by about 0.1 eV/atom from 0 to 50 GPa, but then increases with further increase of the pressure (Fig. 1b). Reducing the concentration of He to 50% (Supplementary Figure 1b), the enthalpy of $MgOHe_{0.5}$ does not decrease from the value at ambient pressure (+0.36 eV/atom) up to at least 300 GPa (+0.41 eV/atom).

Now let us investigate the 2:1 binary ionic compounds. $Li_2O$-He contains, in contrast to $MgF_2$, twice as many cations as anions. However, the insertion of He has a very similar effect as in $MgF_2$. While the enthalpy of formation of the $Li_2OHe$ compound does not become negative with respect to $Li_2O$ and pure He at any pressure in the studied range, it does decrease from +0.25 eV/atom at 0 GPa to almost 0 eV at 300 GPa (Fig. 1c). Its $\Delta H$ is almost on the convex hull at all pressures above 100 GPa (see Supplementary Figure 1c), which agrees with the results of Sun et al. [44]. The reaction enthalpies of the stoichiometries $Li_2OHe_{0.5}$ and $Li_2OHe_2$ also decrease with increasing pressure, but both compounds remain unstable at all pressures studied. In contrast to $Li_2O$-He, LiF-He compounds are not stable, and their reaction enthalpy increases with increasing pressure, i.e., pressure disincentivises the insertion of He in LiF lattices (Fig. 1d).

We also tested the reactivity of He with $CaF_2$, which has an anion:cation ratio of 2:1. The interesting feature of this compound is that it is the prototype of the fluorite structure; remember that the electride $Na_2E$ sublattice of $Na_2He$ can be interpreted as the antifluorite structure. For $CaF_2$, a reaction with He does not cause a departure from the fluorite lattice, but results

merely in the insertion of He in the octahedral interstitials of $CaF_2$. The formation enthalpy of $CaF_2He$ with respect to $CaF_2$ + He shows an intriguing behavior (Fig. 1e): at ambient pressure it is unstable, but its formation enthalpy decreases and becomes negative (stable) at a pressure of about 30 GPa. At pressures higher than 50 GPa, the formation enthalpy increases again, becoming unstable at a pressure of about 110 GPa. The presence of He atoms helps stabilize the ionic compound, but only in the intermediate pressure range of 30–110 GPa. Lastly, we find in agreement with Dong et al. that $Na_2He$ becomes stable above 160 GPa and remains thus up to the highest pressure studied (Fig. 1f).

**Structure changes and electronic properties.** Now, let us analyze the trends in the structures of the compounds formed at high pressure. The most notable feature is that the $A_2BHe$ compounds were found to have the same stable structure with $Fm\bar{3}m$ symmetry at all pressures; see Fig. 2a for an example. This is the structure of full-Heusler compounds. It is also identical to the $Na_2He$ structure when the quasiatoms (E) are considered to be the anions. The second lowest enthalpy structure usually had a symmetry group of $Cmcm$. Its enthalpy was about 0.6 eV/atom higher than the full-Heusler structure throughout the pressure range considered. As in the antifluorite structure, the B ions form an face-centered cubic (FCC) lattice, while the A ions occupy all the tetrahedral sites. This structure ensures that the first neighbor of any ion will be an ion of the opposite charge. The He atoms are inserted into the octahedral sites, thus also forming an FCC lattice. The $A_2B$ compounds also share similar structures at low pressure: $Li_2O$ and $CaF_2$ adopt the same $CaF_2$-type structure at ambient pressure, and $MgF_2$ takes up the $TiO_2$ structure[50,51]. However, these structures have large interstices, making for an inefficient packing, and they will not be thermodynamically favored under very high pressure. As pressure increases all three $A_2B$ ionic compounds adopt more tightly packed structures where the distance between the closest A-B ions and A-A ions are nearly the same (see Supplementary Table 1, and Supplementary Figure 2).

It is interesting that the A-BHe compounds also adopt the same high symmetry structure throughout the pressure range

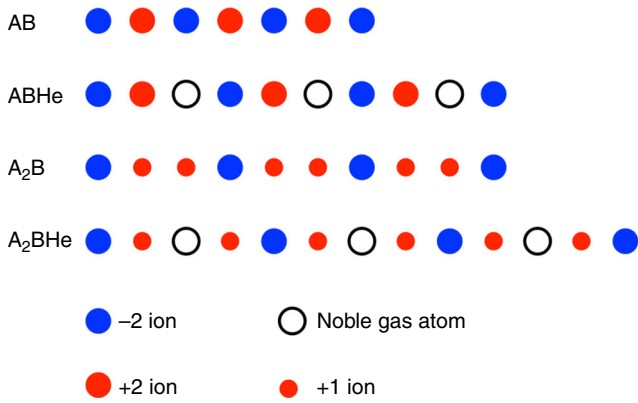

**Fig. 4** One-dimensional schematics of He insertion in AB and $A_2B$ types of ionic compounds. The large red and blue filled circles represent the ions with +2 and −2 charges; the small red filled circles represent the ions with +1 charge; the white circles represent the neutral helium atoms

from 50 to 300 GPa, although the compounds are not stable. Both MgOHe and LiFHe form a structure with $P6_3/mmc$ symmetry. In this structure, shown in Fig. 2d, He atoms occupy a simple hexagonal lattice, while Mg and O occupy hcp lattices. The combination of He and either Mg or O forms a NiAs structure. The Mg and O atoms form an open structure that can accommodate linear chains of He atoms. Both Mg and O atoms have a coordination number of 5.

By studying the electronic structures of these compounds, we can quantitatively examine whether He forms any chemical bonds with the neighboring atoms and species in these inclusion compounds. First, we calculate the electronic localization function (ELF), shown as cross sections in Fig. 3. ELF values close to 1 indicate a high probability of a fully occupied electronic state, such as a filled electronic shell or a covalent bond. As we can see in Fig. 3a, b for both $MgF_2He$ and MgOHe, the ELF has localized, distorted spherical shells around all atoms that are separated by regions of near-zero ELF. The lack of any local ELF maxima away from the atomic sites means that no covalent bonds form between He and the other atoms, nor between Mg and F in $MgF_2He$, and Mg and O in MgOHe. The latter is expected, as the interactions between $Mg^{2+}$ and $F^-$, and $Mg^{2+}$ and $O^{2-}$ are dominantly ionic. A topological analysis of the charge distribution in both compounds[52] confirms this: at 300 GPa, the calculated Bader partial charges on Mg/F and Mg/O in $MgF_2He$ and MgOHe are +1.71/−0.83 and +1.64/−1.56, respectively; the He atoms in both compounds are essentially neutral (0.04 for $MgF_2He$ and 0.07 for MgOHe, respectively; Fig. 3). The major change to the chemical bonding upon insertion of He into the $MgF_2$ and MgO lattices is the change of ionic interactions, in other words Madelung energies, which will be discussed further below.

The inertness of He in these He-salt compounds can also be demonstrated through the electronic projected density of states (PDOS). We calculate and compare three PDOSs for the $MgF_2$ and $MgF_2He$ compounds. First, we obtain the PDOS of Mg-$s/p$, F−$s/p$, and He-$s$ states in $MgF_2He$ at 300 GPa. Second, we obtain the PDOS of Mg-$s/p$ and F-$s/p$ states in a contrived $MgF_2$ compound in which Mg and F atoms occupy the same positions as in $MgF_2He$ at the same pressure. We denote this compound as $MgF_2$[He]. Third, we obtain the PDOS of Mg-$s/p$ and F-$s/p$ states in $MgF_2$ in its most stable structure (*Pnma* symmetry) at 300 GPa. The highest valence bands of all three compounds (Fig. 3c–e) are dominated by the F-$2p$ states of approximately the same width (8–10 eV), and all exhibit very large bandgaps. The He-$1s$ states are mostly located at −15 to −10 eV, but also to some degree around −3 eV, which could just be part of the F-$2p$

states due to overlap of the atom-centered projection spheres. Most importantly, however, after removing the He atoms from $MgF_2He$ but keeping the structure unchanged ($MgF_2$[He]; Fig. 3d) the F-$2p$ states are almost unchanged. This implies that the interaction between He and other atoms in $MgF_2He$ is very small, and there is no hybridization and no chemical bond formation.

More detailed discussions of the effects of He insertions on the electronic and atomic structures of ionic compounds can be found in the Supplementary Information (see Supplementary Notes 2 and 3 as well as Supplementary Figures 2 and 3).

**The driving force of He insertion.** Now we will focus on the mechanism of why stable He + ionic compounds form under pressure. The key issue is why He forms stable compounds with 1:2 (or 2:1) ionic compounds but not with 1:1 compounds. The reason for this can be more easily explained using an example in one spatial dimension. In Fig. 4, we present a very simple, one-dimensional (1D) representation of ionic crystals. The figure shows that in a 1D ionic compound with cation:anion ratio of 1:1 (AB type), the cations and anions are arranged in an alternating fashion; for fixed atomic separation (determined also by the repulsive interactions among atoms in real materials), this is the state with the lowest Madelung energy. If such a compound forms a mixture with NG atoms, the average distance between A and B must increase, increasing the Madelung energy. As a result, the products of AB-type compounds and NG elements will be less stable than the separated phases. On the other hand, for 1D ionic compounds with 2:1 ratio ($A_2B$ type), the ground state contains units of A-B-A (here, we set A as +1 positive-charged and B as −2 negative-charged ions) that repeat infinitely. At the interface of two A-B-A units we will have two A atoms repelling each other. Thus, when NG atoms are inserted in between two A ions, the distance between these two A ions increases, which lowers the Madelung energy, making the structure more stable. The 1D ionic chain model based purely on Coulomb interactions can be solved analytically (see the Supplementary Note 4 and Supplementary Figure 4) and confirms that the insertion of He in $A_2B$-type compounds will lower the Madelung energy, whereas the insertion in AB-type compounds will raise the Madelung energy.

As revealed by the density functional theory calculations and the subsequent analysis of the electronic and structural properties of real He-inclusion materials, it is suggestive that the essence of the mechanism of their stabilization is a modification of electrostatic interactions, i.e., the change of the Madelung energy. This theory is revealed clearly by the simple 1D picture just introduced. However, when discussing the stability of real three-dimensional (3D) materials under pressure, many other factors need to be considered, which will somewhat obscure the above simple argument. Obviously, the effect of the insertion of helium is much smaller in 3D materials because the interstitial sites are naturally larger. Interestingly, both He-inserted $AB_2$ and AB types of ionic compounds show high symmetry lines in their structures (Fig. 2c, f) with the same pattern as we show in Fig. 4.

In order to study the effect of the insertion of He atoms in ionic compound lattices, we discuss separately the two enthalpy contributions of PV work and internal energy, i.e., $H = E + PV$. We then monitor the changes $\Delta E$ and $\Delta(PV)$ upon the insertion reaction, i.e. between the constituents and the product compound. We calculate and plot the two terms as functions of pressure in Fig. 5 for all compounds considered (see more compounds in Supplementary Note 5 and Supplementary Figure 6). It is obvious that $\Delta(PV)$ is zero at ambient conditions ($P = 0$). For reactions involving $AB_2$ or $A_2B$ compounds, $\Delta(PV)$ quickly drops to significantly negative values as a function of pressure. It becomes about −0.2 eV/formula unit for $Li_2O$ and

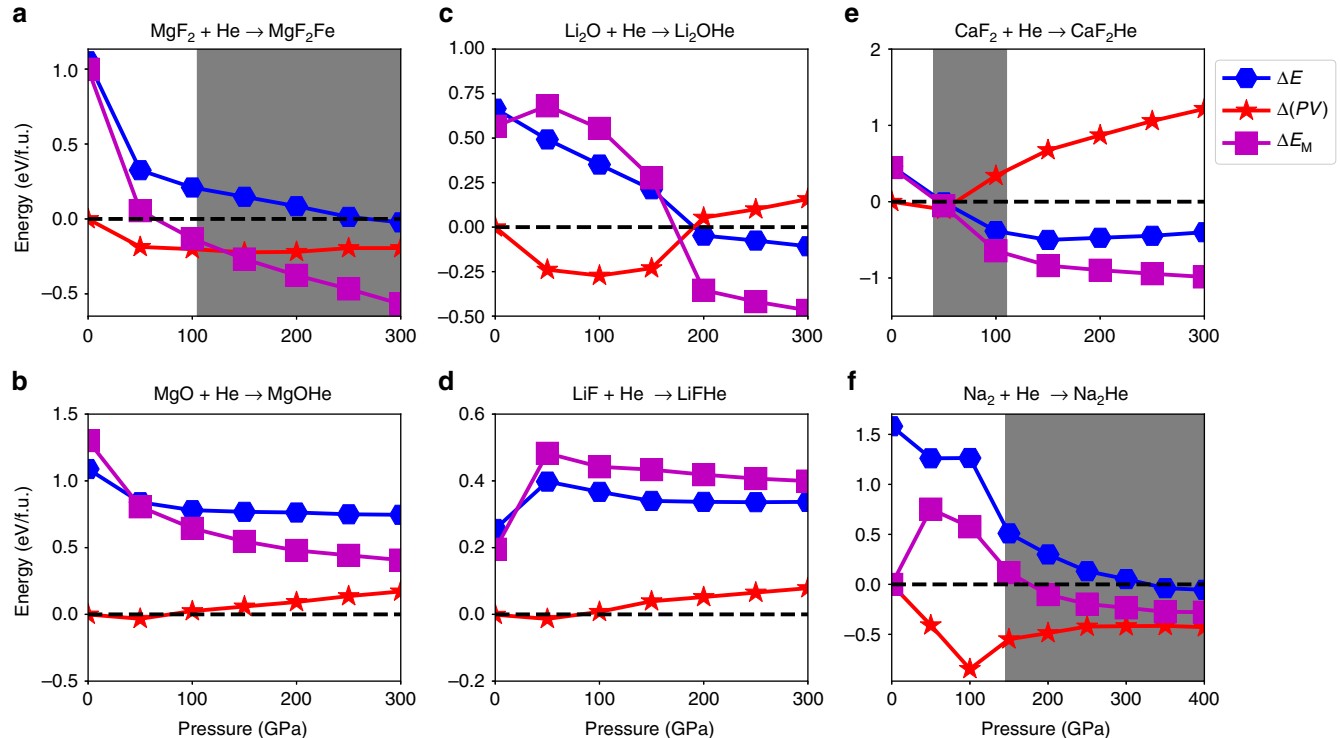

**Fig. 5** Relative changes in $PV$ work, internal energy $E$, and Madelung energy $E_M$ for He insertion. **a–e** The corresponding data of helium inclusion into **a** $MgF_2$, **b** MgO, **c** LiF, **d** $Li_2O$, **e** $CaF_2$, and **f** Na. Relative changes in $PV$ work, internal energy $E$, and numerically determined Madelung energies $E_M$ changes per formula unit are shown with red star, blue hexagonal, and square purple solid lines, respectively

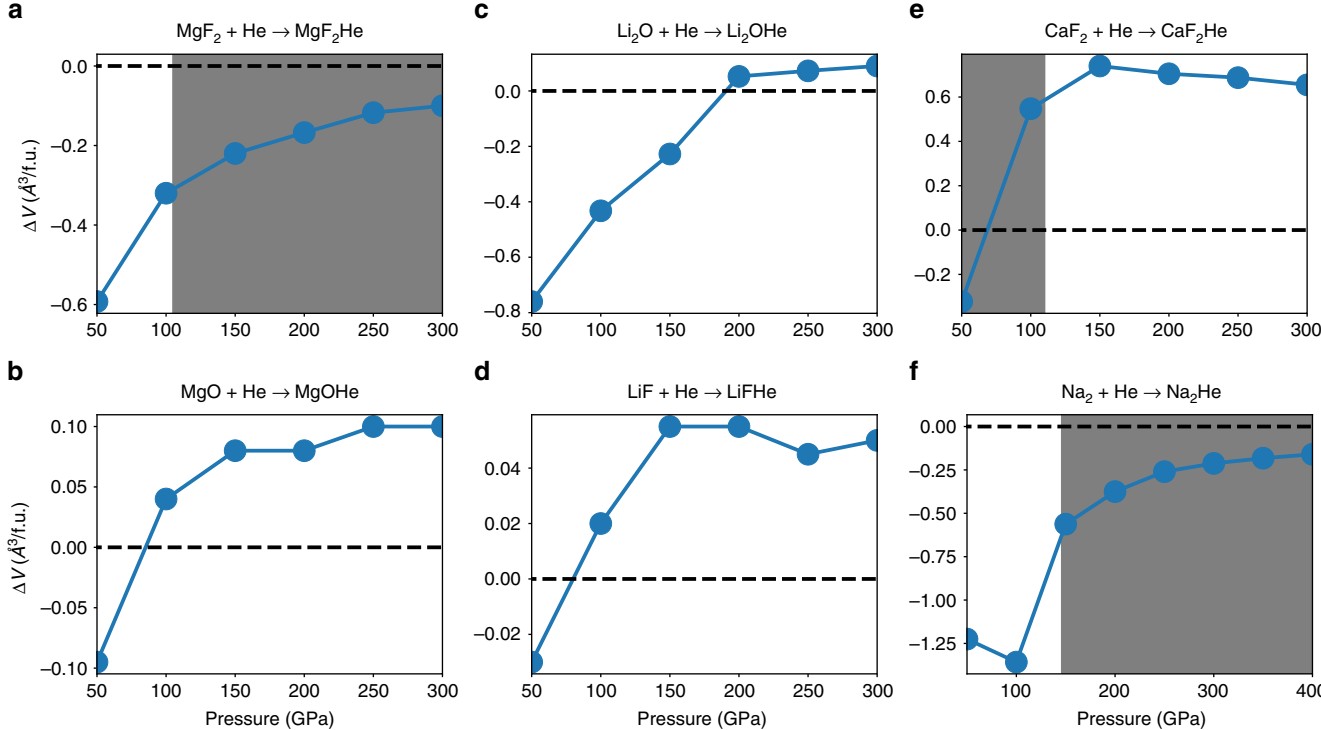

**Fig. 6** The changes of the volume as a function of pressure for He insertion. **a** $MgF_2$, **b** MgO, **c** LiF, **d** $Li_2O$, **e** $CaF_2$, and **f** Na. The dashed lines refer to a sum of the volume of the ionic compounds and elemental helium, whereas the solid line is for the ternary compound. Shaded areas present the pressure intervals of stable He insertion

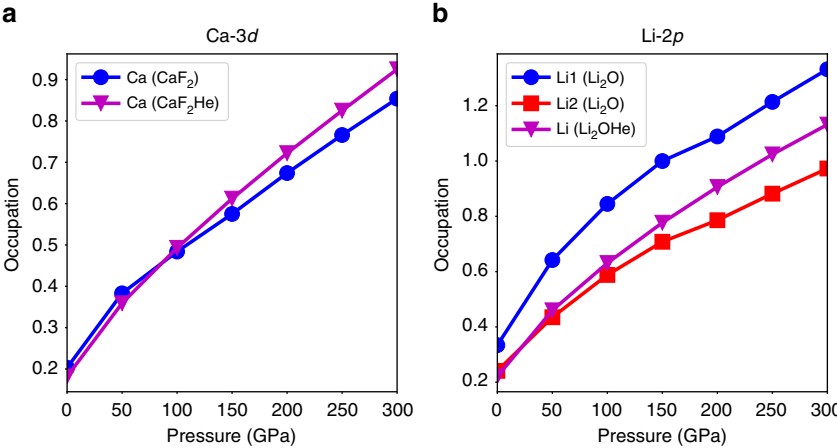

**Fig. 7** Orbital occupations of Ca-3$d$ and Li-2$p$ orbitals in A-B and A-BHe compounds. **a** The $d$ orbital occupation of Ca ions in both CaF$_2$ and CaF$_2$He. **b** The $p$ orbital occupation of Li ions in both Li$_2$O and Li$_2$OHe. Li$_1$ and Li$_2$ represent Li atoms at two inequivalent sites in the crystal

MgF$_2$ compounds and −0.5 eV/formula unit for Na$_2$E beyond 50 GPa. CaF$_2$ is an exception, with $\Delta(PV)$ slightly lower than zero at 50 GPa and positive at higher pressure. In contrast to AB$_2$-type compounds, the value of $\Delta(PV)$ for AB compounds is mostly positive, except for a slightly negative value at low pressure (50 GPa).

The different behaviors of $\Delta(PV)$ are caused by the different volume changes for A$_2$B and AB compounds during the reaction with He. This volume change $\Delta V$ is summarized in Fig. 6. It shows that the insertion of He into the lattice of both A$_2$B and AB types of compounds reduces the overall volume at low pressure, i.e., $\Delta V < 0$; it is advantageous (purely from a $PV$ work perspective) to store helium inside the compounds instead of as separate constituents. However, the volume reduction is much more significant for A$_2$B type of compounds. At ambient pressure, $\Delta V$/formula unit is −0.6 and −0.75 Å$^3$ for MgF$_2$ and Li$_2$O reacting with He, respectively. In contrast, $\Delta V$ is only about −0.1 Å$^3$ for MgO and -0.03 Å$^3$ LiF reacting with He. This distinct difference between A$_2$B and AB types of compounds originates ultimately from the different balance of Coulomb interactions (Madelung energies) of the two types of compounds. As illustrated in the 1D model above, there is strong A-A repulsion in A$_2$B compounds. As a result, A$_2$B compounds assume larger volumes per atom at low pressure to minimize these repulsions, thereby leaving more room for the insertion of He in their lattices. However, the He-inclusion compounds all seem less compressible than the constituents: for both A$_2$B and AB compounds, $\Delta V$ increases with increasing pressure and eventually, for MgO, Li$_2$O, LiF, and CaF$_2$, becomes positive at sufficiently high pressure. That means that the He-inserted lattice has a larger volume than the separate constituent ionic compound and He.

In contrast to the $\Delta(PV)$ term, the insertion of He in the lattice of both AB- and A$_2$B-type compounds causes large increases of the internal energies at ambient and low pressures, $\Delta E > 0$ (Fig. 6). This is due to the disturbance of the electronic structure of the ionic compounds caused by insertion of the NG element. At lower pressure, the gain in $\Delta(PV)$ is not large enough to overcome the large increase of internal energy upon insertion of He. Therefore, at lower pressure, He cannot react with ionic compounds regardless of the cation:anion ratio.

Under increasing pressure, the internal energy balance $\Delta E$ for He insertion decreases significantly. Although this is generally true for both AB and A$_2$B ionic compounds, the decrease of $\Delta E$ is more remarkable for the latter (Fig. 5). For example, $\Delta E$ changes from 1.05 eV/formula unit at 0 GPa to −0.02 eV/formula unit at 300 GPa for MgF$_2$; whereas it only changes from 1.08 eV/

formula unit at 0 GPa to 0.75 eV/formula unit at 300 GPa for MgO. A similar trend can also be found in the Li-based compounds, except that $\Delta E$ actually increases in the pressure range from 0 to 50 GPa for LiF. Because $\Delta(PV)$ either changes only slightly or increases with increasing pressure, it is indeed the dramatic decrease of the internal energy change $\Delta E$ that eventually leads to the stabilization of A$_2$BHe compounds at sufficiently high pressure.

What causes this decrease of $\Delta E$ upon He insertion? One major factor is the change of the Madelung energy as explained in detail for the 1D model. That change of the Madelung energy $\Delta E_M$ can be calculated by assigning effective charges to each atom in the crystals of both the pure ionic compound and the He-inclusion compound. The Bader charges are used as the effective charges for the ions. The results for $\Delta E_M$, for all compounds and pressures, are also shown in Fig. 5. It is obvious that in general $\Delta E_M$ behaves very similar to $\Delta E$ under increasing pressure. The correlation between $\Delta E_M$ and $\Delta E$ indicates that the drastic decrease of the latter under pressure is indeed caused by the change of the Madelung energy. The only major exception occurs in the low-pressure region of Na$_2$He. This is not surprising because Na is not an electride at lower pressure (<200 GPa). Interestingly, Na can form a stable compound with He at 150 GPa, before Na itself becomes quasi-ionic. This can be explained by the theory based on the electrostatic interaction because there is a strong interplay between the electride state and the He insertion. As He is inserted into the Na lattice, it will increase the size of the interstitial sites. Therefore, the quantum orbital energy at the interstitial sites will be lowered, which will help the formation of an electride[53]. In turn, the large local charges in any electride phase will stabilize the insertion of He in the lattice.

The correlation of $\Delta E_M$ and $\Delta E$ is not perfect even for compounds consisting of very hard ions. There are several reasons for this. First, it is hard to truly determine the effective charge of anion in the compounds. The nominal charges are integer numbers and they are usually much larger than the actual charges and the Bader charges. As a matter of fact, these charges may also change with pressure (see the Supplementary Figure 5). However, reasonable variations of the charge values, e.g., by using different calculation methods do not alter the conclusions drawn here. Second, a simple spherical charge distribution model may not work very well for ionic compounds, especially under pressure (notice the non-spherical ELF isosurfaces in Fig. 3a, b). Third, there might be large contributions to the internal energy beyond the Madelung energy. The insertion of He in the ionic compounds increases their lattice constants while at the

same time also blocking the respective interstitial area for other electrons' wavefunctions. The overall effect may raise or lower the kinetic energy of the electrons of the filled anion shells and further influence the internal energy. Lastly, for ionic compounds consisting of heavier ions, such as $CaF_2$, the change of the internal energy may have a turning point and again increase with pressure, opposite to the trend of the Madelung energy. This counteracting factor will be discussed in detail below.

**Opposing factors to He insertion**. In this section, we will examine the question why He-inserted $AB_2$ or $A_2B$ ionic compounds are sometimes not stable even though the reaction potential from the Madelung energy is already significant. For instance, as shown in the previous section, the reaction enthalpy of $He + Li_2O$ decreases with increasing pressure but never becomes negative. Although the Madelung energy and the internal energy both decrease with increasing pressure while He is inserted into the $Li_2O$ lattice, they never form stable compounds. Furthermore, $CaF_2$ forms a stable compound with He but only in a limited pressure range from 30 to 110 GPa. In this case, higher pressure destabilizes the He-inserted ionic compound. Such behavior is also shown in an earlier work of Sun et al. for a number of alkali chalcogenides. For example, we find $K_2S$ to form a stable compound with He in the pressure range from 1.5 to 6.1 GPa (1.3 to 5.8 GPa in the work of Sun et al.[44]).

We will first investigate the unusual behavior of He insertion into the $CaF_2$ lattice. Foremost, its volume change $\Delta V$ increases dramatically with increasing pressure (Fig. 6e). Therefore, although its Madelung energy evolution would stabilize the He insertion, the overall formation enthalpy starts to increase at pressures beyond 50 GPa and the He insertion is not favored at any pressure. This distinct behavior compared to the other ionic compounds is due to the fact that the energy of the Ca-3d orbital is lowered under high pressure, and it becomes partially occupied. This alters dramatically the simple picture of He insertion into this ionic compound. As shown in Fig. 7a, the occupation of the Ca-3d orbital increases from about 0.1 at ambient pressure to 0.3 or 0.4 at 100 GPa. Correspondingly, the charge transfer from Ca to F decreases. As a matter of fact, the Bader charge of Ca in both the $CaF_2$ and $CaF_2He$ compounds decreases from about 1.65 e at 0 GPa to about 1.45 e at 300 GPa. In comparison, the Bader charge of Mg (in MgO and MgOHe) changes much less, from about 1.76 e at 0 GPa to 1.73 e at 300 GPa. A significant charge transfer from F to Ca-3d orbitals leads to greatly reduced repulsive interactions among $F^-$ anions, which lowers the volume of $CaF_2$ under pressure. The overall effect is $\Delta V > 0$ for the helium insertion reaction, which is thus disfavored under pressure. Furthermore, the occupation of the 3d orbitals can lower the kinetic energy under high pressure, because the 3d orbital can largely penetrate into the core region due to the lack of core states with the same angular momentum. This gain in kinetic energy is more significant for $CaF_2$ than $CaF_2He$ since the previous compound is more closely packed. This explains why the internal energy difference $\Delta E$ increases slightly while the pressure is higher than 150 GPa, opposed to the decreasing trend of $\Delta E_M$. The different behavior of $CaF_2$ illustrates that the insertion of He into ionic compounds might be complicated by other factors if the composite species are heavily polarized.

Similar effects arising from the occupation of an orbital with higher angular momentum can also be seen in the $Li_2O$ compound. The Li atom has an electron configuration of $1s^2 2s^1$. However, under high pressure, some of the electrons will be transferred into the 2p orbital. As shown in Fig. 7b, the occupancy of the 2p orbital increases from about 0.2 or 0.3 at 0

GPa to 0.9 or 1.3 at 300 GPa. Due to the lack of any lower shell p orbital, the 2p orbital has no radial node and can largely penetrate into the core region. This essentially reduces the size of the Li ions, which eventually leads to the positive $\Delta V$ in Fig. 6c. Furthermore, our proposed He insertion mechanism and the opposing factors are readily applied to many other $A_2B$ or $AB_2$ compounds as well as Ne insertions in ionic compounds. Several examples are discussed in the Supplementary Information (see Supplementary Notes 5 and 6, as well as Supplementary Figures 6, 7, and 8).

In summary, we propose that chemically inert elements such as He have a prevalent propensity to react with ionic compounds that have unequal numbers of cations and anions. The He atoms do not form any chemical bonds with the ions in the compounds. However, the insertion of He atoms will lower the otherwise strong repulsive Coulomb interactions between the majority ions with the same charge, and therefore lower the Madelung energy. We also show that the recently discovered reactivity of He with Na originates from the same energetic driving force.

## Methods

**Structure search**. In order to test our hypothesis that the insertion of He atoms can lower the Madelung energy of certain types of ionic compounds, we selected a number of compounds with different cation:anion ratios: $Li_2O$ (2:1); LiF (1:1); $MgF_2$ (1:2); MgO (1:1); and $CaF_2$ (1:2), as the test compounds reacting with He. Extensive crystal structure searches were conducted by use of the particle swarm optimization algorithm implemented in CALYPSO (Crystal structure AnaLYsis by Particle Swarm Optimization)[54–57]. A series of efficiency-improving techniques available in the code were employed, including symmetry constraints, bond characterization matrix, and coordination characterization function, etc. The effectiveness and the efficiency of this crystal search method have been proven by numerous early calculations. With the aid of this powerful method, we obtained the predicted stable structures of the above selected ionic compounds and the products of reactions between them and helium. We selected a pressure interval from 0 to 300 GPa and 100 GPa pressure steps for the structure predictions.

**Formation enthalpy and electronic structure calculation**. The formation enthalpy and electronic properties of products were calculated by DFT as implemented in the VASP[58] package, in which the generalized gradient approximation within the framework of Perdew-Burke-Ernzerhof[59] describes the exchange-correlation functional and the projector augmented wave method[60,61] was used to describe electron-ion interactions. For Li (Na, Mg, Ca), the 1s (2s) states were included in the valence. The plane wave cutoff energy is set as 900 eV. The k-point meshes with interval smaller than $2\pi \times 0.05/\text{Å}$ was used for the ab initio calculation and the enthalpies are converged within 1 meV/atom.

**Madelung energy calculation**. The Madelung energy was calculated using a Fourier method that is implemented in the Vesta[62] program. There are two important parameters, including the radius of ion spheres and the Fourier coefficient cutoff frequency. The charge-density distribution, $\rho(r)$, of an ion is defined inside a sphere as $\rho(r) = \rho_0 \left[1 - 6\left(\frac{r}{s}\right)^2 + 8\left(\frac{r}{s}\right)^3 - 3\left(\frac{r}{s}\right)^3\right]$ for $r < s$ else $\rho(r) = 0$, where $s$ is the radius of the sphere. The sphere has to be smaller than half of the interatomic distances. It is determined by testing the convergence of the Madelung energy, a standard procedure as recommended by the VESTA program. The Fourier coefficient cutoff frequency for the long-range Coulomb potential is set as $2/\text{Å}$ for all the calculations. This is also a value recommended by the program.

**Bader charge calculation**. The calculation of the electron population was performed using the Bader Charge Analysis code developed by the Henkelman group in the University of Texas at Austin[63]. While calculating Bader charges, we found that the charges on the He atoms, although very small, are not exactly zero. We would like to point out that this does not mean there is actual charge transfer during the insertion of He into ionic compounds. The He-1s orbital is fully occupied and the 2s orbital is much higher in energy. Therefore, there is no quantum orbital available for any electron transfer. However, while one calculates charges using the Bader analysis, the charge enclosures around He atoms are determined by the zero flux sheets of the total charge density. Even if He atoms form no bonds with the surrounding atoms, the total charge density is the overlap of He electrons and the electrons of neighboring ions. Therefore, the enclosed charge around He might be slightly different from 2. The Bader charge of He in $Na_2He$ is even higher because the charge in the interstitial sites (quasiatoms) overlaps more with the He atoms.

**Data availability**. The data supporting this publication are available from the authors on request.

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

## Acknowledgements

We acknowledge partial financial support from NSAF U1530401 and computational resources from the Beijing Computational Science Research Center. Z.L. and D.Y. acknowledge the National Natural Science Foundation of China (NSFC) for grants under Nos. 21374011 and 21434001. E.Z. acknowledges the NSF (DMR-1505817) for financial support. A.H. acknowledges the Royal Society (RG-150247) for financial support. Most of the calculations are performed on NSF-funded XSEDE resources (TG-DMR130005) especially on the Stampede cluster run by Texas Advanced Computing Center.

## Author contributions

M.S.M. proposed the mechanism and designed the study. Z.L., J.B., and M.S.M. conducted most of the calculations. M.S.M., D.Y., and H.L. coordinated and guided the research. All authors were involved in data analysis and result discussions. S.V. contributed to the Madelung energy analysis. M.S.M, Z.L., J.B., A.H., E.Z., and H.L. wrote and revised the manuscript together.

## Additional information

**Competing interests:** The authors declare no competing interests.

