## [Peer Review File · Nature Communications]

Reviewers' comments:

Reviewer #1 (Remarks to the Author):

The paper proposes a general reaction propensity of Helium with ionic compounds that contain unequal number of cations and anions under high pressure. This greatly extends the range of reactivity of Helium and probably also other noble gas elements.

Using extensive calculations and 1D analytical models, the paper demonstrates that the compelling force of the reactivity is not forming any local chemical bonds but rather the change of the long range Coulomb interactions between the ions due to the insertion of He atoms. The same mechanism is then used to explain the recently discovered reaction of Helium with Sodium under high pressure.

Because most of the Earth minerals, including Earth mantle materials, consist of unequal number of cations and anions, the current work suggests that there might be a considerable amount of helium being stored in the Earth mantle.

I recommend the publication of the paper in case that the following points have been amended:

- 1) The paper argues that the formation of stable Na_2He can be explained by the reaction of helium with ionic compounds since Na becomes electrides under high enough pressure. However, Na_2He are predicted to form at a pressure before Na becomes electride.
- 2) As shown by the calculations, Li_2O does not form stable compound with He under pressures up to 300 GPa.
- 3) The Madelung energy calculation depends on the choice of the effective charges of ions. The paper didn't discuss how does the different choice affect the eventual results and why they choose to use the Bader charge.
- 4) Although the Madelung energy shows a change under pressure in parallel with that of internal energy, it is not clear why there is a net difference between the two. Where does this energy difference come from, and why does it change only slightly with pressure?
- 5) Ref. 45 should be replaced by the citation to the CALYPSO code of Wang et al., *Comput. Phys. Commun.* 183, 2063 (2012).
- 6) There is a referee repetition on Refs. 18 and 33.

Reviewer #2 (Remarks to the Author):

This manuscript presents and provides support for a suggestion that the stability of He containing ionic compounds can be understood from a straightforward and fairly simple intuitive arguments. The authors suggest, and provide computational support, that the stability of He containing ionic compounds essentially requires an unequal number of cations and anions. Chemical bonds between He and the ionic components are suggested as not being required or the essential features for stability in these materials. Electrostatic energy arguments employing Madelung

energies provide essential supporting information for their suggestion. The ideas are supported by convex hull calculations employing density functional theory (DFT) calculations together with state-of-the-art structure search methods.

- 1) The use of the term "react" when they do not form chemical bonds may be a little misleading. Perhaps simply indicating the "formation of stable insertion structures" may be more consistent with the authors' intentions.
- 2) Are these to be considered simply as inclusion compounds? Are any of these to be considered as electrides in detail? The discussion includes comments on electrides, in particular for the Na-He material but it is probably useful to include more detail although the answer to these questions is somewhat outside the main theme of this manuscript. In particular, have electrides been considered for the other compounds considered in the manuscript?
- 3) There is some reported charge transfer in the Na₂He and He-H₂O compounds at high pressures. The authors may want to just mention this for completeness. It will not detract from their main argument. The authors mention this feature in some of the compounds on page 6 but this charge transfer is somewhat larger in Na₂He.
- 4) A scale needs to be added on the ELF plots in Figure 3a and 3b to indicate at least the maximum and minimum ELF values in the figure.
- 5) The authors may also have bandgap versus pressure information that could be useful along with their enthalpy of reaction information. In other words, are the compounds becoming more insulating at higher pressure?
- 6) Were the structures selected the lowest energy structures found with the CALYPSO code or were they selected for their simplicity?

As a summary, this manuscript provides an interesting simple idea to suggest a main feature for the stability of structures formed with ionic materials and He. Simple ideas are often the best ideas. The idea has been supported with DFT calculations using reliable codes. These results should definitely encourage further experiments (and theory) in the search for possible stable structures with He. Publication is supported after modifications are made as indicated in the comments.

Reviewer #3 (Remarks to the Author):

This manuscript 'Chemistry without Chemical Bonds: Reactivity of He with Ionic Compounds under High Pressure' by Liu et al., provides an interesting explanation to the formation of ionic compounds involving atomic helium at extreme pressure conditions. Finding the conditions where the most inert element helium can start to form compounds with other elements has been an interesting question in high pressure science. Other inert gases have been shown to form compounds at ambient and high pressures decades ago, so it seemed to be a good question to ask whether helium with only two electrons could also form a solid with interactions going beyond those of van der Waals solids. Although the theoretical prediction has been out for several years, not until recently the stable compounds (Na₂He and Na₂OHe) consisting of helium have been reported. This manuscript by Liu et al. provides a theoretical explanation on why Na₂He and Na₂OHe are able to form, and also goes beyond this point to predict a series of new He compounds that may be stabilized by external applied pressure as well. Basically, the authors suggest that helium atoms do not form any chemical bonding with other elements; they only act as physical spacers in the solid matrix, and stabilize the compound by reducing the long range Coulomb repulsions (lower the Madelung energy). I think this is an interesting idea that is potentially publishable in Nature Communications, considering the recent excitement of finding first He compounds. However, this study lapses into several considerations and discussion which make the argument flawed. These problems should be adequately addressed. Specially,

(1) The authors conclude that He only form compounds with 1:2 or 2:1 ionic compounds but not with 1:1 compounds. But, this conclusion is based on the examination of only 5 compounds (as shown in Fig. 1). I do not think such a small sample pool can adequately support this conclusion.

(2) The proposed '1 D model for stabilization' is flawed and misleading. One has to keep in mind there are *always* both attractive and repulsive forces between an anion and a cation in a solid – so that they do not collapse to each other. The balance of these two forces determine the equilibrium pair distance. The Madelung energy is the sum of the attractive terms ($-e^2M/4\pi\epsilon r$) only (as shown in the equation on p.9). Using Madelung energy alone as a measure of chain stability, without taking into account the repulsive term ($\sim B/r^m$), is incorrect. One can already see this error, for example, in Fig. 5b. Any elongation or reduction from equilibrium bond distance should result in an energy increase, but this figure shows when r increase from $1.0 r_1$ to $1.2 r_1$ and $1.4 r_1$ the energy reduces. This therefore indicates the linear chain should fall apart – there is no potential-energy-well to hold the structure. I hate to say this: I know the authors feature this model prominently and formulate their main argument around it but it is inadequate. I think they should either build a correct model or drop this part completely, and reformulate a new discussion on the structures (correctly).

(3) This theory does not explain the mechanism of helium forming compound with covalent matrix. Could this be explained as well?

Reply to Reviewers' comments:

#####

Reviewer #1 (Remarks to the Author):

#####

The paper proposes a general reaction propensity of Helium with ionic compounds that contain unequal number of cations and anions under high pressure. This greatly extends the range of reactivity of Helium and probably also other noble gas elements.

Using extensive calculations and 1D analytical models, the paper demonstrates that the compelling force of the reactivity is not forming any local chemical bonds but rather the change of the long range Coulomb interactions between the ions due to the insertion of He atoms. The same mechanism is then used to explain the recently discovered reaction of Helium with Sodium under high pressure.

Because most of the Earth minerals, including Earth mantle materials, consist of unequal number of cations and anions, the current work suggests that there might be a considerable amount of helium being stored in the Earth mantle.

I recommend the publication of the paper in case that the following points have been amended:

1) The paper argues that the formation of stable Na_2He can be explained by the reaction of helium with ionic compounds since Na becomes electrified under high enough pressure. However, Na_2He are predicted to form at a pressure before Na becomes electrified.

Reply: This is a very important point that has not been explicitly explained in the manuscript. It is absolutely true that a stable Na_2He compound can already form at a pressure that Na is not yet an electrified. However, there is a strong interplay between the electrified state and the He insertion. While

He is inserted into the Na lattice, it will increase the size of the interstitial sites. Therefore, the quantum orbital energy at the interstitial sites will be lowered, which will help the formation of the electronegative state. In turn, the large charges in the electronegative will stabilize the insertion of He in the lattice. We have explained this in detail in the new version.

2) As shown by the calculations, Li₂O does not form stable compound with He under pressures up to 300 GPa.

Reply: This is again a very important point that hasn't been emphasized enough in the previous version. What we propose is a new driving force for the insertion of He atoms (and other noble gases) in ionic compounds. Such a driving force can be clearly seen for the insertion of He into Li₂O. As a matter of fact, the reaction enthalpy of Li₂OHe decreases continuously up to 150 GPa, at which point it is almost zero, indicating that Li₂OHe may well be made as a metastable compound. However, it is interesting to notice that the reaction enthalpy starts to increase with increasing pressure beyond 150 GPa.

The overall stability of the He inserted compounds depends on the balance of the above driving force with other factors. One important factor is the volume change during such reactions. At lower pressures, the total volume of A₂B (or AB₂) + He usually decreases while He is inserted. This is due to the fact that the large ion-ion repulsive interactions in A₂B compounds increase their volumes, leaving large room for the He insertion. However, with increasing pressure, the close packing becomes more important. When the pressure is high enough, the separate A₂B (or AB₂) ionic compound and elemental He will have a smaller volume compared with the ternary compound, because it is hard to attain close packing for ions and atoms of different sizes. This effect is intensified in Li₂O because the occupation of the Li 2p orbitals increases under high pressure. The Li-2p orbital has no radial nodes and can penetrate extensively into the core region, which will largely reduce the volume of the Li ion. This will greatly enhance the close packing, leading to an unfavorable volume increase for He insertion.

After analyzing the energy terms in a number of He insertion reactions, we found that the above intricate balance of factors influencing the stability of Li₂OHe is present in many cases. This includes CaF₂He, which becomes stable in the pressure range from 50 GPa to 100 GPa, but is destabilized at higher pressure. Similar to Li₂OHe, the factor opposing the Madelung energy reduction is the occupation of the Ca 3d orbitals. Many more examples are now given in the expanded Supplementary Information.

In summary, the insertion of He into A₂B (or AB₂) ionic compounds always exhibits a reduction of the Madelung energy, which is the driving force to form stable compounds. However, the eventual stability depends on other factors including the change of the volume, which is influenced by the occupation of orbitals with a higher angular momentum. In the current version, we add a new section on the important opposing factors and discuss their effects in detail.

3) The Madelung energy calculation depends on the choice of the effective charges of ions. The paper didn't discuss how does the different choice affect the eventual results and why they choose to use the Bader charge.

Reply: We agree with the referee that the Madelung energy calculation depends on the choice of the effective charges of the ions. The nominal charges are integer numbers and they are usually much larger than the actual charges. There are many ways to define charges. We choose the Bader charge because it is well defined and is ready for use together with VASP calculations. In contrast to Mulliken charges for example, the Bader charges of a compound add up to exactly zero, and they do not depend on any pre-defined projection sphere radii.

Furthermore, we would like to emphasize that the general trend revealed by our calculations, *i.e.* the reduction of the Madelung energy due to He insertion, does not depend on the methodology used to set charge values for the Madelung energy calculations. If the nominal charges are used, the Madelung energies are overestimated, and their reduction is amplified. We found that the trend of the Madelung energy change (ΔE_M) using the nominal charge compares well with those calculated from the Bader charge. Please see and compare the figure below with Fig. 6 in the paper. We have added a discussion at the end of the section on the driving force to reiterate this point.

4) Although the Madelung energy shows a change under pressure in parallel with that of internal energy, it is not clear why there is a net difference between the two. Where does this energy difference come from, and why does it change only slightly with pressure?

Reply: The internal energy consists of many terms besides the Madelung energy, including for example the electronic exchange-correlation energy, electronic kinetic energy etc. These terms are clearly affected by pressure, however the insertion of He into the lattice typically has a much smaller influence on these terms as compared to its effect on the Madelung energy. This is again because the insertion of He into ionic compounds does not form or break chemical bonds, but rather it modifies the distance between negatively and positively charged ions in the lattice.

However, we do find, in some cases, that the insertion of He has an important effect not only on the Madelung energy. In these cases, the change of the internal energy ΔE and the change of the Madelung energy ΔE_M do not evolve in parallel with increasing pressure. As a matter of fact, in several cases such as CaF_2 and K_2S etc, ΔE actually increases beyond a certain pressure, opposing the trend of decreasing ΔE_M . This happens in compounds where the electrons will occupy the cations' higher-level d orbitals under pressure. These d levels can largely penetrate into the core region and therefore lower the kinetic energy. A detailed discussion of this point has been included in a new section in the paper.

5) Ref. 45 should be replaced by the citation to the CALYPSO code of Wang et al., *Comput. Phys. Commun.* 183, 2063 (2012).

Reply: We take the advice of the referee and have changed the reference.

6) There is a referee repetition on Refs. 18 and 33.

Reply: We thank the referee for noticing this error and have corrected it.

#####

Reviewer #2 (Remarks to the Author):

#####

This manuscript presents and provides support for a suggestion that the stability of He containing ionic compounds can be understood from a straightforward and fairly simple intuitive arguments. The

authors suggest, and provide computational support, that the stability of He containing ionic compounds essentially requires an unequal number of cations and anions. Chemical bonds between He and the ionic components are suggested as not being required or the essential features for stability in these materials. Electrostatic energy arguments employing Madelung energies provide essential supporting information for their suggestion. The ideas are supported by convex hull calculations employing density functional theory (DFT) calculations together with state-of-the-art structure search methods.

1) The use of the term “react” when they do not form chemical bonds may be a little misleading. Perhaps simply indicating the “formation of stable insertion structures” may be more consistent with the authors’ intentions.

Reply: We understand the referee’s concern and to a large extent agree with it. The formation of He inserted ionic compounds is very different to any chemical reactions we have known before. That is why this chemical transformation is so interesting. On the other hand, the formation of a new and **stable** compound from different substances is in general a process that matches the definition of a chemical reaction. We took the advice of the referee and changed the title. However, the word reaction has been used in so many places in the manuscript, and it is really hard to replace them all.

2) Are these to be considered simply as inclusion compounds? Are any of these to be considered as electrides in detail? The discussion includes comments on electrides, in particular for the Na-He material but it is probably useful to include more detail although the answer to these questions is somewhat outside the main theme of this manuscript. In particular, have electrides been considered for the other compounds considered in the manuscript?

Reply: This is an important point and we are sorry that it is not explicitly shown in the earlier version. All the new examples presented in this work (i.e. beyond Na₂He) including MgF₂, MgO, Li₂O, LiF and CaF₂ are simple ionic compounds and their reaction with He can be considered a “simple” insertion. It is indeed a very important point that the insertion of He and the formation of an electride in Na are two interplaying processes. The insertion of He in the Na lattice increases the size of the interstitial sites and therefore helps the formation of an electride. A similar effect has been seen in other work, such as the insertion of Xe in an Mg lattice helps to form electride materials for Mg at a very low pressure (~ 125GPa). [*Miao et al. JACS, 137, 14122 (2015)*] In turn, the formation of the electride also helps the insertion of He because of the lowering of the Madelung energy. Due to the importance of this point, we have revised the corresponding part of the manuscript and thoroughly discussed it. (see the second last paragraph in the section of “driving force”)

3) There is some reported charge transfer in the Na₂He and He_H₂O compounds at high pressures. The authors may want to just mention this for completeness. It will not detract from their main argument. The authors mention this feature in some of the compounds on page 6 but this charge transfer is somewhat larger in Na₂He.

Reply: We take the advice of the referee and have thoroughly discussed the issue of the non-zero Bader charges of He in the Methods section of the manuscript. We would like to point out that this does not mean there is actual charge transfer during the insertion of He into the ionic compound. The He 1s orbital is fully occupied and the 2s orbital is much higher in energy. Therefore there is no quantum orbital available for any electron transfer. However, while one calculates charge using a Bader analysis, the charge enclosures around He atoms are determined from the maxima of the total charge density. Even if He atoms have zero bonding with the surrounding atoms, the total charge density is the overlap of He electrons and the electrons of neighboring ions. Therefore, the enclosed charge around He may differ slightly from 2. The Bader charge of He in Na₂He is even higher because the charge in the interstitial sites (quasi-atoms) overlaps more with the He atoms.

4) A scale needs to be added on the ELF plots in Figure 3a and 3b to indicate at least the maximum and minimum ELF values in the figure.

Reply: We thank the referee for pointing out the omission of the scale in ELF plot. We have corrected this in the new version.

5) The authors may also have bandgap versus pressure information that could be useful along with their enthalpy of reaction information. In other words, are the compounds becoming more insulating at higher pressure?

Reply: We take the advice of the referee and have studied the change of the band gaps as a function of pressure for Mg compounds and the corresponding He inserted compounds. We find interesting features and have added new panels in Figure 4 and provide an extensive discussion of the results. We thank the referee for the nice suggestion.

6) Were the structures selected the lowest energy structures found with the CALYPSO code or were they selected for their simplicity?

Reply: Yes, all the structures are the most stable structures found from structure search using CALYPSO. We took several low enthalpy structures from the search and calculated their enthalpy as a function of pressure. For all of the He inserted A₂B or AB₂ example compounds, the full-Heusler structure remains the most stable one throughout the pressure range studied. As a matter of fact, the second lowest enthalpy structure usually has a symmetry group of Cmcm. Throughout the pressure range, it is about 0.6 eV/atom higher in enthalpy than the full-Heusler structure!

As a summary, this manuscript provides an interesting simple idea to suggest a main feature for the stability of structures formed with ionic materials and He. Simple ideas are often the best ideas. The idea has been supported with DFT calculations using reliable codes. These results should definitely encourage further experiments (and theory) in the search for possible stable structures with He. Publication is supported after modifications are made as indicated in the comments.

#####

Reviewer #3 (Remarks to the Author):

#####

This manuscript 'Chemistry without Chemical Bonds: Reactivity of He with Ionic Compounds under High Pressure' by Liu et al., provides an interesting explanation to the formation of ionic compounds involving atomic helium at extreme pressure conditions. Finding the conditions where the most inert element helium can start to form compounds with other elements has been an interesting question in high pressure science. Other inert gases have been shown to form compounds at ambient and high pressures decades ago, so it seemed to be a good question to ask whether helium with only two electrons could also form a solid with interactions going beyond those of van der Waals solids. Although the theoretical prediction has been out for several years, not until recently the stable compounds (Na₂He and Na₂OHe) consisting of helium have been reported. This manuscript by Liu et al. provides a theoretical explanation on why Na₂He and Na₂OHe are able to form, and also goes beyond this point to predict a series of new He compounds that may be stabilized by external applied pressure as well. Basically, the authors suggest that helium atoms do not form any chemical bonding with other elements; they only act as physical spacers in the solid matrix, and stabilize the compound by reducing the long range Coulomb repulsions (lower the Madelung energy). I think this is an interesting idea that is potentially publishable in Nature Communications, considering the recent excitement of finding first He compounds. However, this study lapses into several considerations and discussion which make the argument flawed. These problems should be adequately addressed. Specially,

(1) The authors conclude that He only form compounds with 1:2 or 2:1 ionic compounds but not with 1:1 compounds. But, this conclusion is based on the examination of only 5 compounds (as shown in Fig. 1). I do not think such a small sample pool can adequately support this conclusion.

Reply: We understand the referee's concern. Actually, there are many more compounds showing this behavior. We choose five compounds as examples to demonstrate our theory, because both cations and anions in them are hard ions with very low polarization. The major change to these compounds

when the He atoms are inserted is the Madelung energy. Besides these five compounds and Na₂He, Na₂O has been shown to form He inserted compounds under pressure in the original paper describing Na₂He. Sun et al. have conducted a systematic study of the formation of He (and Ne) inserted compounds with alkali oxides and sulfides, although no clear mechanism has been provided in that work. [Sun et al. arXiv 1409.2227]

In our expanded Supplementary Materials, we analyze the terms influencing compound formation with He, where we provide the Madelung energy for 8 compounds including the oxides and sulfides of Na, K, Rb and Cs. Our analysis clearly reveals that the reduction of the Madelung energy is a driving force for the insertion of He in these ionic compounds. However, due to the increased occupation of outer shell d orbitals in K, Rb and Cs under pressure, some of the He inserted compounds become unstable at higher pressure (please see the Manuscript and Supplementary materials for details). Furthermore, we also show in the supplementary materials that Na₂O and K₂S can form stable compounds with Neon in the proper pressure range. They show very similar behavior while compared with the He inserted ionic compounds, and their stability also originates from the reduction of the Madelung energy that occurs upon the insertion of Ne into the ionic crystal lattice.

In summary, we would emphasize that the driving force of He (and other noble gas atoms) insertion is ubiquitous. However, the overall stability of the Ng-containing compound depends on the balance of this driving force with other factors. Ionic compounds consisting of light elements can show this effect more exclusively. In the new version, we added a section to thoroughly discuss the major factors opposing Ng insertion, namely the occupation of outer shell orbitals that can largely penetrate into the core region.

(2) The proposed '1 D model for stabilization' is flawed and misleading. One has to keep in mind there are always both attractive and repulsive forces between an anion and a cation in a solid – so that they do not collapse to each other. The balance of these two forces determine the equilibrium pair distance. The Madelung energy is the sum of the attractive terms ($-e^2M/4\pi\epsilon r$) only (as shown in the equation on p.9). Using Madelung energy alone as a measure of chain stability, without taking into account the repulsive term ($\sim B/r^m$), is incorrect. One can already see this error, for example, in Fig. 5b. Any elongation or reduction from equilibrium bond distance should result in an energy increase, but this figure shows when r_2 increase from 1.0 r_1 to 1.2 r_1 and 1.4 r_1 the energy reduces. This therefore indicates the linear chain should fall apart – there is no potential-energy-well to hold the structure. I hate to say this: I know the authors

feature this model prominently and formulate their main argument around it but it is inadequate. I think they should either build a correct model or drop this part completely, and reformulate a new discussion on the structures (correctly).

Reply: We thank the referee for raising this point. We employed this model because it helps us to present the basic idea in a simple way. First, we totally agree with the referee about the incompleteness of the model. We were fully aware of the instability of the one-dimensional ionic

point charges from the beginning. Samuel Earnshaw, a British mathematician, proved that a set of point charges can never be maintained in a stable form purely by electrostatic interactions. As suggested by the referee, adding local repulsive interactions and an external stress can stabilize the 1D ionic chain in a stable structure. However, our purpose is to illustrate how the Madelung energy *changes* when inserting neutral atoms in between charged ions. The true demonstration of the stability of He insertion can only be done for real 3D compounds in which many other geometric and energetic factors (including for example the local repulsive interactions) can be included. They are fully treated at the quantum mechanics level by our electronic structure calculations based on the DFT method.

Originally, we choose this 1D model for two reasons. First, it clearly illustrates the simple idea that the Madelung energy will be reduced by inserting neutral atoms, which is the proposed driving force of forming the He inserted compounds. Secondly, an analytic solution for the Madelung energy can be obtained for this model. We would like to emphasize that the analytical solution of the Madelung energy is not necessary to demonstrate the idea. In fact, the simple schematic is good enough to show that the insertion of He will definitely reduce the Madelung energy for A_2B or AB_2 type of ionic compounds. Since the analytical solution is not the critical part, we decided to move it to Supplementary Information.

(3) This theory does not explain the mechanism of helium forming compound with covalent matrix. Could this be explained as well?

Reply: Examples of He insertion into covalent compounds are very rare. A good example is the insertion of He into silica (SiO_2) either in glassy form or in the cristobalite structure. Either polymorph contains large interstitial voids that can accommodate He atoms. It was found by DAC experiment that both silica glass and cristobalite can transform into two new phases in Helium media [Sato *et al. Nat. Comm.* 2, 345 (2011); Shen *et al., PNAS*, 108, 6004 (2011), Sato *et al. Phys. Chem. Mineral.* 40, 3 (2013)]. However, the structure measurement cannot determine the presence and the sites of the He atoms. A follow-up DFT calculation showed that two He inserted structures corroborating the above XRD patterns are indeed lower in enthalpy than the unreacted cristobalite and He. It also found that there is only a very narrow window (5 – 25 GPa) that He inserted silica is stable [Matsui *et al, Am. Mineral.* 99, 184 (2014)]. We repeated the above calculations and confirmed this result. Furthermore, we found that if a more stable form of silica is used, the He inserted silica can barely be stabilized by pressure at about 5 GPa.

[Redacted]

Considering that the SiO_2 system is very different from the ionic compounds that are studied in this work, we decided not to include a discussion of it in this paper (other than a very brief mention in the concluding remarks). [Redacted]

We thank the referee for raising this question.

REVIEWERS' COMMENTS:

Reviewer #1 (Remarks to the Author):

The revision is satisfactory and I recommend the publication as is.

Reviewer #2 (Remarks to the Author):

This manuscript has been revised in response to the comments of all reviewers. The revised manuscript presents a clear description that supports an interesting idea on the basis for the stability of He insertion compounds formed with ionic solids. The current manuscript includes an expanded critical discussion of the basic idea that the stability of He-ionic compounds can be understood from a picture based on a Madelung energy analysis. The authors provide interesting and reasonable arguments for the stability of A₂B compounds as compared to that of AB compounds. The authors have both addressed the comments and have made clear additions following the suggestions of reviewers that have helped the authors make an even better manuscript. I have only a few very minor additional comments for the authors regarding the manuscript and Supplementary Information that do not affect the content of the manuscript.

1) The units of charge should perhaps be added to Figure S3.

2) I do not see some of the figures mentioned in the text of the Supplementary Information section.

3) The K-point meshes used could be mentioned in the Methods section. Were they all the same?

In Summary, the revised manuscript is a clearly written account of an idea regarding the stability of He-Ionic compounds and I now feel it contains sufficient critical discussion and additional material. The interesting idea proposed and critically discussed will almost certainly stimulate additional experiments and theoretical analysis.

Reviewer #3 (Remarks to the Author):

The authors have done a laudable job addressing my questions and the other two reviewers' questions. A number of the errors have been amended and confusion clarified. In my opinion, this paper is publishable in Nature Communications now.

The response to the referees:

Reviewer #1 (Remarks to the Author):

Comments:

The revision is satisfactory and I recommend the publication as is.

Reply:

We thank Referee's comment on our paper.

Reviewer #2 (Remarks to the Author):

This manuscript has been revised in response to the comments of all reviewers. The revised manuscript presents a clear description that supports an interesting idea on the basis for the stability of He insertion compounds formed with ionic solids. The current manuscript includes an expanded critical discussion of the basic idea that the stability of He-ionic compounds can be understood from a picture based on a Madelung energy analysis. The authors provide interesting and reasonable arguments for the stability of A₂B compounds as compared to that of AB compounds. The authors have both addressed the comments and have made clear additions following the suggestions of reviewers that have helped the authors make an even better manuscript. I have only a few very minor additional comments for the authors regarding the manuscript and Supplementary Information that do not affect the content of the manuscript.

Comment 1: *The units of charge should perhaps be added to Figure S3.*

Reply: We add the charge unit e in Supplementary Fig. S3

Comment 2: *I do not see some of the figures mentioned in the text of the Supplementary Information section.*

Reply: We didn't find the mentioned figure in supplementary material that was not cited in the manuscript. In addition, we changed the figures in the supplementary materials name according to the format requirement to make the figures clearer to find.

***Comment 3:** The K-point meshes used could be mentioned in the Methods section. Were they all the same?*

Reply: This is a very important point we need to add to the method. We thank Referee for pointing this out. The number of k-points depends on the size of the unit cell. Therefore, instead, we specify the grid interval of the k-mesh that is $2\pi \times 0.05 \text{ \AA}^{-1}$. This fine k-mesh ensures all the calculations converge to less than 1 meV per atom.

In Summary, the revised manuscript is a clearly written account of an idea regarding the stability of He-Ionic compounds and I now feel it contains sufficient critical discussion and additional material. The interesting idea proposed and critically discussed will almost certainly stimulate additional experiments and theoretical analysis.

Reviewer #3 (Remarks to the Author):

Comments:

The authors have done a laudable job addressing my questions and the other two reviewers' questions. A number of the errors have been amended and confusion clarified. In my opinion, this paper is publishable in Nature Communications now.

Reply:

We thank Referee's comments and suggestions on our paper.